# Bypassing Emergency Service: Decoding the Drivers of Self-Referral During Acute Myocardial Infarction on Rural Areas in Sachsen-Anhalt, Germany

**DOI:** 10.3390/healthcare12222234

**Published:** 2024-11-09

**Authors:** Karen Holland, Sara L. Lueckmann, Mohamad Assaf, Rafael Mikolajczyk

**Affiliations:** Faculty of Medicine, Institute of Medical Epidemiology, Biometrics, and Informatics, Interdisciplinary Center for Health Sciences, Martin Luther University Halle-Wittenberg, 06112 Halle Saale, Germany; karen.holland@uk-halle.de (K.H.); sara.lueckmann@uk-halle.de (S.L.L.);

**Keywords:** acute myocardial infarction, ambulance, self-transport, self-referral

## Abstract

**Background/Objectives:** the timely and effective management of acute myocardial infarction (AMI) is crucial to improve patient outcomes. ‘Self-Referral’ is defined as instances either where patients arrive at the hospital by their own means or are transported by someone else, rather than through professional emergency medical services. This approach can lead to treatment delays and potentially worsen outcomes. This study aims to identify the factors associated with the choice of self-referral among patients with AMI in Saxony-Anhalt, Germany. **Methods:** We used the data from the Regional Myocardial Infarction Registry of Saxony-Anhalt (RHESA), which included 4044 patients with confirmed acute myocardial infarction (AMI), including 48.7% from urban areas (city of Halle) and 51.3% from rural areas (Altmark). The gender distribution was 65% male and 35% female, covering an age range from 25 to over 80 years. Multivariable logistic regression identified factors associated with self-referral and its impact on reaching a hospital with percutaneous coronary intervention (PCI) capability. **Results:** Rural residents were more likely to self-refer compared to those in urban settings (adjusted odds ratio 2.43 [95% CI: 2.00–2.94]). Odds of self-referral decreased with age, while metabolic factors, including hypertension, high body mass index (BMI), and diabetes, as well as sex were not associated with self-referral. Self-referral did not increase the odds of arriving in a hospital without PCI capability. (Adjusted odds ratio 1.12 [95% CI: 0.85–1.47]). Furthermore, in cases of self-referral, women did not have a disadvantage in reaching a hospital with PCI (0.91; 0.59–1.41) compared to men. However, in medically attended transports, women were at a disadvantage (odds ratio: 1.33; 95% CI: 1.06–1.67). **Conclusions:** These findings highlight the need for public education on self-referral and for medical personnel training to prevent gender bias in AMI transport to PCI-capable hospitals.

## 1. Introduction

Cardiovascular diseases, prominently acute myocardial infarction (AMI), are a leading cause of global mortality [1,2]. In Germany, mortality and morbidity rates for AMI vary strongly across regions. Saxony-Anhalt has a high age-standardized mortality rate (67 per 100,000) compared to the national average (46 deaths per 100,000 inhabitants) [3,4,5,6]. Multiple factors such as healthcare infrastructure, socioeconomic status, and individual risk profiles may contribute to this disparity. However, comprehensive understanding of the underlying causes is still lacking [3,4,7]. Moreover, rural and urban settings often differ significantly in terms of healthcare accessibility, infrastructure, and patient behavior, all of which can affect the management and outcomes of acute coronary syndromes (ACSs). Rural areas, like many parts of Saxony-Anhalt, may face longer EMS response times, fewer hospitals equipped with percutaneous coronary intervention (PCI) capabilities, and stronger reliance on family physicians, potentially leading to delays in receiving optimal care. In contrast, urban areas typically have shorter response times and greater access to specialized facilities, which may lead to better outcomes in AMI management. The timely identification of symptoms and initiation of evidence-based therapies [8,9], particularly reperfusion strategies such as percutaneous coronary intervention (PCI) and thrombolysis, play a critical role in minimizing myocardial damage and improving survival rates in AMI cases [10,11,12,13]. Leading medical organizations, such as the American Heart Association (AHA) and the European Society of Cardiology (ESC), emphasize the necessity of adhering to recognized care standards to ensure timely and efficacious AMI management [12,14,15].

By adhering to evidence-based recommendations, healthcare professionals can ensure that they provide the most appropriate and rapid care, which could enhance several patient outcomes. Furthermore, the patient’s mode of arrival at the hospital, whether through self-referral or ambulance transportation, directly dictates the speed at which they receive medical intervention and the suitability of the treatment, both crucial determinants of survival [16,17,18].

When dealing with AMI, the choice of self-referral has notable implications. Patients who decide to independently seek hospital care, bypassing emergency medical services (EMSs), risk potential delays [16,18,19]. Contrastingly, ambulance transportation provides swift access to medical care, enhanced by the benefit of in-transit paramedic support. Paramedics can deliver initial medical care and are skilled in recognizing and providing urgent treatment for Acute Coronary Syndrome, including the management of fatal complications, such as arrhythmias and cardiac arrest. This Intervention ensures appropriate patient care and a smoother transition to hospital-based treatment [19,20]. Moreover, ambulances facilitate directing patients to hospitals with requisite facilities and expertise [21,22], such as specialized cardiac care units, catheterization labs for immediate angioplasty procedures, and experienced cardiology teams proficient in emergency interventional procedures.

Extensive research on ambulance usage has shown varying utilization rates among AMI patients, ranging from 23% to 60% [16,17,18,19,20,23,24]. Factors contributing to this variation include symptom perception, which is influenced by public education and the severity of symptoms, as well as symptom misinterpretation [17,19,20]. Other determinants encompass a history of heart attacks, gender, age, race, insurance status, and distance from the hospital [16,24].

Self-referral behavior, particularly in the European population, has not been sufficiently studied. The primary objective of this study is to examine self-referral among AMI patients in two regions of Saxony-Anhalt and to identify demographic and metabolic factors associated with it. The secondary objective was to examine the impact of self-referral to facilities that are lacking adequately equipped cardiac units.

## 2. Materials and Methods

### 2.1. Study Population and Data Collection

This analysis used data from the Regional Myocardial Infarction Registry of Saxony-Anhalt (RHESA), which includes a sample of 4044 individuals with myocardial infarction. The register was described elsewhere [25]. In brief, the RHESA includes individuals who had a diagnosis of either non-ST segment elevation myocardial infarction (NSTEMI) or ST segment elevation myocardial infarction (STEMI) confirmed by electrocardiogram, aged 25 years and older, residing in two regions in Saxony-Anhalt, Germany—city of Halle (urban) and Altmark (rural). Altmark is a rural region in northern Saxony-Anhalt, characterized by small towns, low population density, and limited healthcare infrastructure, primarily focused on agriculture.

The information on AMI patients was obtained through the Hospital Data Collection Form, which is a questionnaire completed by medical professionals in hospitals. Recruitment for the RHESA and its baseline information were obtained between June 2013 and December 2019. All patients admitted with a myocardial infarction diagnosis were included in the initial population. For this analysis, we excluded patients with an unknown transport mode and information about the initial hospital of admission (missing data) and those who were already hospitalized at the moment of the infarction. Data on educational grade, social level, and marital status were excluded due to a high number of missing values.

### 2.2. Definition of Independent Variables

We investigated various demographic and clinical factors that could potentially be linked to self-referral, such as age, sex, geographic region, and comorbidities. The pre-existing risk factors and comorbidities included diabetes mellitus, arterial hypertension, and hypercholesterolemia. We calculated body mass index (BMI) as weight in kilograms divided by height in meters squared. Adhering to World Health Organization (WHO) standards, we categorized participants into five BMI groups: ‘Normal or below’ (merging underweight and normal weight, <25), ‘Overweight’ from 25 to 30, ‘Obesity I’ from 30 to 35, ‘Obesity II’ from 35 to 40, and ‘Obesity III’ > 40. The age of the patients at the time of their myocardial infarction was divided into five age groups (25–45, 50–59, 60–69, 70–79, 80+). Other collected variables included sex and residential status, categorized as rural or urban.

### 2.3. Outcomes

The primary outcome of this study was self-referral to a hospital in connection with AMI. We defined self-referral as arrival by private means, such as a personal vehicle or public transport, or being transported by third parties. In contrast, non-self-referral included cases where patients were referred from a medical facility, such as general hospitals, emergency departments, or by the emergency medical services. This non-self-referral category encompasses referrals made by ‘Emergency Services transport’ (including emergency physician vehicles, emergency ambulances, and helicopters) and referrals made by ‘Family Physicians’ in outpatient settings, which encompasses the following categories: ‘Emergency Services’ (emergency physician vehicles, emergency ambulances, and helicopters) and ‘Family Physician’. The secondary outcome was defined as “PCI Capability”, indicating whether the initial hospital of arrival following a cardiac event was equipped with a percutaneous coronary intervention (PCI) unit.

### 2.4. Statistical Analysis

The descriptive results are presented as mean ± SD for continuous variables and absolute and relative frequencies with 95% confidence intervals for categorical variables. The margin of error for these estimates was calculated using standard methods based on sample size and data variability, ensuring a 95% confidence interval for all reported proportions and odds ratios.

In order to address missing values in the dataset, we employed the Multiple Imputation by Chained Equation (MICE) method [26], under the assumption that the data were Missing At Random (MAR). Demographic and metabolic predictors including age, sex, region, hypertension, BMI categories, diabetes mellitus, and hypercholesterolemia were included in the process. Our analysis involved generating five imputed datasets via the MICE algorithm and employing predictive mean matching for imputation. The imputed datasets were subsequently utilized in further analyses, with the imputation process’s variability duly incorporated. Finally, we verified the imputed data’s plausibility by comparing its distribution with the original dataset to ensure no unexpected patterns emerged.

We first described the three groups with respect to transport mode: Emergency Service, Family Physician, and Self-Referral. Subsequently, we dichotomized the main outcome variable for logistic regression, into Self-Referral or Not Self-Referral.

Simple logistic regressions were performed to assess the association of each metabolic and demographic characteristic with the outcome. In a subsequent analysis, we included variables selected based on their clinical relevance and statistical associations in the multivariable logistic regression model. This model incorporated demographic variables such as age, region, and sex, as well as metabolic factors. Our second outcome was whether the hospital to which the patient arrived had the PCI capability.

We calculated odds ratios (ORs) and the corresponding 95% confidence intervals (CIs). All statistical analyses were performed using R 4.3.1 software. Ethical approval for RHESA was obtained from the Ethics Committee of the Medical Faculty at Martin Luther University (reference number: [2013-32]).

## 3. Results

### 3.1. Sample Characteristics

A total of 4044 patients with AMI were included in the analysis. Of these, 65.3% were male. The average age of the study population was 69.5 years. Among the sample, 85.1% of patients had hypertension, 52.9% had hypercholesterolemia, and 35.4% diabetes. In terms of BMI, 40.8% of patients were classified as overweight, and 32.9% were classified as obese. Within the obese category, Type I obesity was the most predominant, representing 22.4% of the sample. General characteristics, encompassing demographic and metabolic parameters, are shown in Appendix A (see Appendix A), while Table 1 presents the distribution of key characteristics between self-referral and non-self-referral groups within the rural population, highlighting differences in age, sex, and metabolic conditions.

### 3.2. Variables Associated with Self-Referral Behavior

After adjusting for all considered variables in the multivariable model, self-referral was more common in the rural compared to the urban area. Diabetic patients were more inclined to opt for visiting a family physician as their primary contact (Table 1). Also, younger age groups were more likely to use self-referral compared to older (Table 2).

### 3.3. Self-Referral to Hospitals Lacking PCI Capabilities

In the general sample, patients who self-referred were 1.82 times more likely to present at hospitals lacking percutaneous coronary intervention (PCI) capabilities compared to their non-self-referred counterparts (initial unadjusted odds ratio, 95% CI: 1.53–2.16). Given that all urban hospitals were equipped with PCI facilities, the analysis was subsequently adjusted to focus exclusively on rural regions, where 63.4% patients with AMI arrived in hospitals without PCI capability. At the same time, the use of self-referral as a mode of transportation to the hospital in the rural region of Altmark was observed in 23.3% of cases, while other modes of transportation combined accounted for 76.7%.

In the rural sample, the multivariable analysis showed that self-referral was not associated with arrival at a hospital without PCI capability compared to non-self-referral. However, women were more often admitted to hospitals without PCI capability [Table 3].

Looking further into gender disparity through the stratified comparative analysis, we found that among the self-referral population, there was no noticeable difference between females and males in terms of arriving at hospitals without PCI capability. However, women who did not self-refer were more frequently directed to hospitals lacking PCI facilities compared to their male counterparts (Table 4).

## 4. Discussion

Despite recommendations from bodies like the AHA and ESC, many patients still opt for self-referral instead of professional medical transport in AMI cases. Sachsen-Anhalt, a region with one of the highest AMI incidence and mortality rates in Germany, has not seen the same improvements in cardiovascular outcomes as other states. Our study, through a regional AMI registry, aims to explore the factors contributing to this persistently high mortality and identify targeted measures for its reduction.

In our sample, self-referral was more common among younger patients, likely due to an underestimation of symptom severity or misinterpretation as another condition [27,28]. Younger individuals may believe self-referral offers quicker access to care, despite studies showing it does not lead to faster medical attention and lacks pre-hospital treatment benefits [23,24]. In contrast, older patients, aware of their comorbidities and higher health risks, tended to use EMSs. In some cases, they may not have had a choice, with healthcare decisions made by others. Self-referral was also more common in rural areas, possibly due to healthcare accessibility and public awareness issues [29,30].

Altmark is a predominantly rural region in Sachsen-Anhalt, Germany, characterized by low population density and limited healthcare infrastructure compared to urban areas. In this setting, Emergency Medical Services (EMSs) are crucial for the initial treatment and transport of patients with acute myocardial infarction (AMI). However, response times for EMSs are often longer due to greater distances and limited resources. Furthermore, many hospitals in the region lack percutaneous coronary intervention (PCI) capabilities, leading some patients to self-refer in an effort to avoid delays. Rural patients, who often have strong relationships with their family physicians, may choose to visit their doctor first rather than going directly to the emergency department. These regional characteristics likely contributed to the self-referral patterns observed in our study. This pattern aligns with urban–rural disparities seen in other studies [31,32]. Rural areas face significant challenges in accessing specialized care, such as PCI, due to fewer hospitals and longer distances to facilities with adequate resources. This “rural gap” makes timely and adequate care harder to achieve, especially in emergencies like AMI. Additionally, factors like lower levels of education and limited health literacy can influence healthcare-seeking behavior and decision making, potentially delaying appropriate care.

In contrast, all urban centers were equipped with specialized facilities capable of performing PCI, with the lack of PCI capability being an issue exclusively in rural regions. In this subset, self-referral did not show a significant impact. However, the likelihood of arriving at a hospital without PCI capabilities increased with age, particularly for individuals aged 80 and above, women, and those with diabetes mellitus. Conversely, hypertension and hypercholesterolemia were associated with lower odds of being taken to an unequipped hospital. One possible explanation is that diabetic patients, women, and older individuals often experience milder or atypical symptoms, along with decreased sensibility [33], making it more difficult for the EMS to assess the severity of their condition.

In descriptive terms, we observed that diabetic patients were more likely to visit their family physician before being referred or transported to the hospital, rather than directly using emergency services. This finding may suggest the influence of ongoing physician–patient relationships in chronic illness care [34,35], though more evidence is needed to fully understand this dynamic. While long-term trust and continuous care with their general practitioner may play a role in this behavior, our data do not allow for definitive conclusions. In the German healthcare system, general practitioners act as “gatekeepers”, coordinating patient care and providing referrals to specialists when necessary. According to guidelines from the American Heart Association, optimal PCI following an AMI must occur within a narrow time window, emphasizing the critical need for timely access to care [36,37]. In our study, a gender gap was observed, with female patients using EMSs having higher odds of being taken to a hospital without PCI capabilities. This finding was not the case when women self-referred. Given the time-sensitive nature of AMI treatment, such delays can result in poorer outcomes [38,39]. The disparity may be related to persisting underestimation or misinterpretation of cardiovascular symptoms in women [40,41,42]. This gender discrepancy highlights the urgent need for a healthcare system that recognizes and addresses gender-based differences to ensure equitable treatment for all patients. It also underscores the broader rural–urban healthcare disparities identified in this study, where self-referral behaviors and access to PCI-equipped facilities vary significantly. Addressing these gaps requires targeted interventions and greater public awareness, particularly in rural areas. These findings suggest potential areas for future research on how public health campaigns and educational initiatives can influence healthcare-seeking behaviors and improve outcomes in acute cardiac care.

## 5. Limitations

One limitation of our study is that the findings are specific to the region, reflecting its local medical resources, healthcare system, and patient behaviors. These results may not be generalizable to other settings with different infrastructures, referral systems, or population characteristics.

## 6. Conclusions

Our study found that younger adults and rural residents were more likely to opt for self-referral in case of AMI symptoms. Comprehending how these factors influence patient decision making is essential for healthcare providers. Given the high percentage of self-referral in these specific groups, targeted educational interventions are crucial. Such efforts could improve timely access to appropriate medical care, particularly in underserved rural areas. Our findings highlight the need for public health campaigns to raise AMI symptom awareness and emphasize the importance of professional medical transport to reduce cardiovascular mortality and morbidity. Unfortunately, the underestimation or misinterpretation of cardiovascular symptoms in women, lead more often to medical transport to hospitals without PCI capability, while self-referral women were not at a disadvantage.

## Figures and Tables

**Table 1 healthcare-12-02234-t001:** Sample characteristics in rural areas, divided by mode of transport (self-referral vs. non-self-referral.

Characteristic	Overall; [95% CI] ^2^	Non-Self-Referral; [95% CI] ^2^	Self-Referral; [95% CI] ^2^
**Sex**			
Male	65.1%; [63.0, 67.2]	63.9%; [61.5, 66.2]	69.2%; [64.9, 73.3]
Female	34.9%; [32.8, 37.0]	36.1%; [33.8, 38.5]	30.8%; [26.7, 35.1]
**Age categories**			
25–49	9.1%; [7.9, 10.4]	7.9%; [6.7, 9.4]	12.8%; [10.0, 16.2]
50–59	20.3%; [18.6, 22.1]	19.2%; [17.3, 21.2]	24.0%; [20.3, 28.1]
60–69	20.2%; [18.5, 22.0]	19.7%; [17.8, 21.8]	21.9%; [18.3, 25.9]
70–79	27.0%; [25.1, 28.9]	26.9%; [24.8, 29.2]	27.1%; [23.2, 31.3]
80+	23.5%; [21.7, 25.4]	26.3%; [24.1, 28.5]	14.3%; [11.3, 17.8]
**Arterial Hypertension**	89.4%; [88.0, 90.7]	90.3%; [88.7, 91.7]	86.4%; [82.9, 89.2]
**Hypercholesterolemia**	75.0%; [73.0, 76.8]	76.7%; [74.5, 78.7]	69.2%; [64.9, 73.3]
**Diabetes**	33.2%; [31.2, 35.3]	34.8%; [32.5, 37.2]	27.9%; [24.0, 32.2]
**BMI**			
(1) Normal_weight or less	23.8%; [22.0, 25.7]	24.3%; [22.2, 26.5]	22.3%; [18.7, 26.3]
(2) Overweight	41.6%; [39.4, 43.7]	41.1%; [38.7, 43.6]	43.0%; [38.5, 47.5]
(3) Obesity I	23.9%; [22.1, 25.8]	23.4%; [21.4, 25.6]	25.6%; [21.8, 29.8]
(4) Obesity II	7.4%; [6.3, 8.6]	7.7%; [6.5, 9.2]	6.2%; [4.3, 8.8]
(5) Obesity III	3.3%; [2.6, 4.2]	3.4%; [2.6, 4.4]	2.9%; [1.7, 4.9]
1%			

^2^ CI = Confidence Interval.

**Table 2 healthcare-12-02234-t002:** Variables associated with self-referral.

Simple Regression Models			Multivariable Logistic Regression Model	
Variable	Odds Ratio (OR) ^1^	95% CI ^1^	Odds Ratio (OR) ^1^	95% CI ^1^
Sex [Female vs. Male]	0.79	0.66–0.94	0.93	0.77–1.12
Age categories ^2^				
50–59	0.76	0.57–1.0	0.76	0.56–1.03
60–69	0.65	0.48–0.87	0.70	0.52–0.95
70–79	0.51	0.39–0.68	0.56	0.42–0.76
[80+]	0.31	0.22–0.42	0.34	0.24–0.47
Region [Rural vs. urban]	2.24	1.89–2.66	2.43	2.00–2.94
Arterial Hypertension [Yes vs. no]	0.80	0.64–0.99	0.85	0.67–1.09
Hypercholesterolemia [Yes vs. no]	1.21	1.03–1.42	0.82	0.68–1.00
Diabetes [Yes vs. no]	0.81	0.68–0.96	0.97	0.81–1.17
BMI ^3^				
Overweight	1.21	0.98–1.49	1.13	0.92–1.40
Obesity	1.24	1.00–1.55	1.14	0.92–1.43

Observations: 4037. ^1^ OR = odds ratio; CI = confidence interval; ^2^ reference group: age group 25–49; ^3^ reference group: BMI Normal weight or less.

**Table 3 healthcare-12-02234-t003:** Multivariable analysis: arriving into non-PCI capability hospital in rural areas.

Predictor	Odds Ratio (OR) ^1^	95% CI ^1^	*p*-Value
Self_referral Yes vs. No ^3^	1.11	0.89, 1.38	0.4
Sex Female vs. Male	1.22	1.00, 1.50	0.053
Age categories ^2^			
50–59	0.85	0.59, 1.22	0.4
60–69	1.23	0.84, 1.78	0.3
70–79	1.28	0.88, 1.84	0.2
80+	2.11	1.43, 3.11	<0.001
Arterial Hypertension Yes vs. No	0.36	0.24, 0.53	<0.001
Hypercholesterolemia Yes vs. No	0.58	0.46, 0.74	<0.001
Diabetes Yes vs. No	1.37	1.12, 1.68	0.002
BMI^4^			
Overweight	1.12	0.88, 1.42	0.3
Obesity	1.10	0.86, 1.41	0.4

^1^ OR = odds ratio, CI = confidence interval, ^2^ reference age category 25–49, ^3^ reference no self-referral, ^4^ reference group: BMI Normal weight or less.

**Table 4 healthcare-12-02234-t004:** Comparative multivariable analysis of self-referral and non-self-referral AMI patients arriving at hospitals without PCI capability in rural areas.

Self-Referral Population				No Self-Referral Population		
Predictor	Odds Ratio (OR) ^1^	95% CI ^1^	*p*-Value	Odds Ratio (OR) ^1^	95% CI ^1^	*p*-Value
Sex Female vs. Male	0.91	0.59, 1.41	0.7	1.33	1.06, 1.67	0.015
Age categories ^2^						
50–59	0.86	0.42, 1.73	0.7	0.88	0.57, 1.36	0.6
60–69	0.891	0.44, 1.86	0.8	1.40	0.90, 2.19	0.13
70–79	0.76	0.37, 1.54	0.5	1.57	1.02, 2.43	0.040
80+	1.77	0.77, 4.12	0.2	2.35	1.50, 3.67	<0.001
Arterial Hypertension Yes vs. No	0.22	0.08, 0.50	<0.001	0.41	0.26, 0.63	<0.001
Hypercholesterolemia Yes vs. No	0.61	0.38, 0.99	0.057	0.58	0.44, 0.77	<0.001
Diabetes Yes vs. No	1.28	0.82, 2.01	0.3	1.41	1.12, 1.78	0.003
BMI ^3^						
Overweight	1.43	0.86, 2.38	0.2	1.07	0.81, 1.40	0.6
Obesity	1.54	0.91, 2.61	0.11	1.01	0.76, 1.34	>0.9

^1^ OR = odds ratio, CI = confidence interval, ^2^ reference age category 25–49, ^3^ reference group: BMI normal weight or less.

## Data Availability

The datasets used and/or analyzed during the current study are available from the corresponding author upon reasonable request.

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
