# Peer review of "Bypassing Emergency Service: Decoding the Drivers of Self-Referral During Acute Myocardial Infarction on Rural Areas in Sachsen-Anhalt, Germany"

_healthcare, 2024, doi:10.3390/healthcare12222234_

Round 1
Reviewer 1 Report
Comments and Suggestions for Authors
Revision Note
Journal: Healthcare (ISSN 2227-9032)
Manuscript ID: healthcare-3175830
Type: Article
Title: Bypassing Emergency Service: Decoding the Drivers of Self-Referral during Acute Myocardial Infarction
1-Regarding the Abstract Section:
1- The purpose of the definition of ‘Self-Referral’ should be clearly explained. The sentence ‘Using 16 data from the Regional Myocardial Infarction Registry of Saxony-Anhalt (RHESA).’ given in the ‘Material-Method’ section of the article does not comply with the spelling rules and is insufficient to explain this section. It is recommended that the sample size, methods used, and patient selection criteria (e.g., age-gender homogeneity, age group, male and female percentage) be explained in the ‘Material-Method’ section given as a whole in the abstract. If there is a significant result of the ‘odds ratios’ given in this section, it is recommended that it be expressed with numerical data.
2-Regarding the Materials and Methods section
2.1. Study population and data collection
1-Study sample group size should be defined. The method used when diagnosing patients with ‘STEMI’ or ‘AMI’ should be clearly stated (single expert opinion, ECG findings? operative procedures? operation recommendations depending on the level of development of the hospital in the specified years?), and the inclusion and exclusion criteria should be clearly stated.
3-Regarding the ‘Discussion’ section
1-There are different articles with similar purposes in the literature; sentences explaining the specific value of the study should be included based on current references, and each statistically significant result detected should be included in the discussion section. The specificity of the article was found insufficient.
Reviewer 2 Report
Comments and Suggestions for Authors
1. Much of the paper's content focuses on a rural area in Germany, which is highly region-specific. It is recommended that the title should reflect this specific research area.
2. As mentioned above, the study's results may be influenced by factors such as the EMS, the emergency department system, and patient behaviors. A separate section should be dedicated to describing these factors in this specific area (Altmark).
3. (Line 104) What institutions or groups are referred to in the text?
4. (Line 124) How were the patients referred by family physicians classified?
5. The healthcare system varies by region, so please clarify whether patients were transferred from an outpatient clinic to the emergency department within the same hospital. Additionally, indicate whether these patients were self-referred or non-self-referred.
6. In Table 1, a high proportion of patients were referred by family physicians. Please explain whether this is related to the local healthcare system.
7. For patients with myocardial infarction, the primary goal of EMS is to stabilize the patient and transport them to a hospital equipped to provide appropriate treatment. However, the study results indicated that self-referral behavior did not increase the risk of reaching a hospital without treatment capabilities. The authors should comprehensively explain this finding. Also, please describe the availability of medical resources in this area.
8. Table 4 shows that in the rural area, non-self-referred patients had a significantly higher risk of being taken to hospitals without PCI capability among women, diabetic patients, and older patients —especially those over 80 years old. However, this risk was not observed in self-referred patients. The authors should explain this.
9. In rural areas, diabetes was identified as a risk factor for non-self-referred patients reaching a hospital without PCI capability. Between lines 214 and 220, the authors mention that in Table 1, diabetes was most common among patients referred by family physicians. The reviewer finds it important to know whether these patients could reach hospitals with adequate PCI capabilities. The authors should analyze the outcomes for patients referred by family physicians to determine whether this group achieves the best outcomes (i.e., arriving at hospitals with PCI capability), supporting the authors' assertion about the importance of a long-term doctor-patient relationship.
10. The study's findings are not limited to gender issues. A more comprehensive discussion of the study's significant findings could provide valuable insights for improving AMI patient care in rural areas.
11. Since the study's results are closely tied to local medical resources, healthcare systems, and patient behaviors in Altmark, the authors should discuss the generalizability of these findings and include this as a limitation in the paper.
Reviewer 3 Report
Comments and Suggestions for Authors
Cardiac infarctions continue to be the primary cause of deaths worldwide, especially in highly industrialized areas. The authors discussed the self-referral aspects of rural and urban cases in their topics. Frankly, I don't know what the contribution to the literature is. Although it is a different article, the authors need to address some deficiencies.
1. In the abstract section, the start and end dates of the study, age, gender and significant findings with real p values ​​should be added in the results.
2. The introduction section can generally be expanded on urban and rural acute coronary syndromes rather than the topics.
3. In the material method, the universal size, margin of error and acceptability status of the cases included in the study and the method they were calculated should be explained.
4. Acceptance and rejection criteria should be explained.
5. Why did the authors prefer patients over the age of 25? Today, the ACS patient age has decreased to under 18, it should be explained. In addition, why was regression analysis not performed for those under the age of 49.
6. When distinguishing between rural and urban cases, the article can be further strengthened with ROC curve analysis in terms of correlation and mortality.
7. Based on the results of the study, the authors should write a few sentences about its contribution to the literature and make it more striking to read.
Best regards.
Comments on the Quality of English LanguageMinor editing of English language required.
Author Response
Please see the attachment. (answers are at the end of document)

Round 2
Reviewer 1 Report
Comments and Suggestions for Authors
As a result of the suggested revisions, the author's effort to correct the article was found to be sufficient at this stage, although not perfect.
Author Response
thanks
Reviewer 2 Report
Comments and Suggestions for Authors
1. The manuscript title was revised to highlight the regional focus of this study. However, the content was not updated to reflect this change. For instance, a new Table 1 should be provided, showing only the rural sample and dividing it into self-referral and non-self-referral groups. Without this, readers may struggle to understand the characteristics of the rural sample in Sachsen-Anhalt.
2. The authors did not fully address my comment No. 9. Without providing sufficient evidence, emphasizing the importance of long-term doctor-patient relationships appears to be an overinterpretation.
3. The discussion section should be shortened and reorganized, as many parts are repetitive and redundant. This section should focus exclusively on issues related to the study's findings. In addition to the behavioral differences between urban and rural patients, the unfavorable outcomes among women, diabetic patients, and older adults are key findings that warrant more in-depth exploration.
4. In line 152, the results can be derived through simple calculations, making adjustments unnecessary.
5. The results in Tables 3 and 4 are very similar. The authors should provide an explanation for this, as it may indicate a potential limitation of the manuscript.
6. The authors are encouraged to resubmit this manuscript after making the necessary revisions.
Author Response
Please see the attachment.
table general Sample is now in appedix

Reviewer 3 Report
Comments and Suggestions for Authors
It is seen that the authors have made many requested changes. The work can be accepted as it is.
Best regards.
Author Response
thank you
Round 3
Reviewer 2 Report
Comments and Suggestions for Authors
Some sections in the revised version remain unchanged, and the discussion section lacks a clear focus. Certain parts of this section do not align with the research findings. Additionally, points 5 and 6 were not sufficiently addressed.
Author Response
Dear Reviewer,
Thank you for your thorough review and the insightful comments on our manuscript. I appreciate the time you took to evaluate our work and your suggestions for further improvement. After careful consideration and consultation with co-authors, we have decided to maintain the current version of the manuscript. We believe it adequately addresses the objectives of the study and stands by the conclusions drawn. We respect your expertise and understand your points. Thank you once again for your valuable input.